# Spatial targeting of Screening + Eave tubes (SET), a house-based malaria control intervention, in Côte d'Ivoire: A geostatistical modelling study

Tiago Canelas[1]*, Edward Thomsen[1], Daniel McDermott[1], Eleanore Sternberg[1], Matthew B. Thomas[2,3], Eve Worrall[1]

1 Vector Biology, Liverpool School of Tropical Medicine, Merseyside, United Kingdom, 2 Department of Entomology and Center for Infectious Disease Dynamics, The Pennsylvania State University, State College, Pennsylvania, United States of America, 3 York Environmental Sustainability Institute, University of York, Yorkshire, United Kingdom

* tiago.canelas@lstmed.ac.uk

**Data Availability Statement:** All datasets used for this work are publicly available or in the additional

## Abstract

New malaria control tools and tailoring interventions to local contexts are needed to reduce the malaria burden and meet global goals. The housing modification, screening plus a targeted house-based insecticide delivery system called the In2Care® Eave Tubes, has been shown to reduce clinical malaria in a large cluster randomised controlled trial. However, the widescale suitability of this approach is unknown. We aimed to predict household suitability and define the most appropriate locations for ground-truthing where Screening + Eave Tubes (SET) could be implemented across Côte d'Ivoire. We classified DHS sampled households into suitable for SET based on the walls and roof materials. We fitted a Bayesian beta-binomial logistic model using the integrated nested Laplace approximation (INLA) to predict suitability of SET and to define priority locations for ground-truthing and to calculate the potential population coverage and costs. Based on currently available data on house type and malaria infection rate, 31% of the total population and 17.5% of the population in areas of high malaria transmission live in areas suitable for SET. The estimated cost of implementing SET in suitable high malaria transmission areas would be $46m ($13m – $108m). Ground-truthing and more studies should be conducted to evaluate the efficacy and feasibility of SET in these settings. The study provides an example of implementing strategies to reflect local socio-economic and epidemiological factors, and move beyond blanket, one-size-fits-all strategies.

## Introduction

To eliminate malaria, the WHO Global Technical Strategy recommends that new malaria control tools that overcome the challenges of insecticide resistance and residual transmission are urgently explored [1]. In addition, WHO's Global Vector Control Response revolves around the ultimate goal of implementing locally adapted tools [2].

material. House materials are in the Demographic Health Survey from USAID. Covariates sources used in the geostatistical model are listed in additional material 3. Coding for the geospatial dataset is available at: https://protect-eu.mimecast.com/s/HfHDCv8yvi7Yw4RhQB7rw?domain=datadryad.org For figures 1, 3, 4, 5 the basemaps were obtained from the Humanitarian Data Exchange: https://data.humdata.org/dataset/cote-d-ivoire-administrative-level-0-3-boundaries-and-points.

**Funding:** This work was supported by the Bill & Melinda Gates Foundation, grant no. OPP1131603, and TC and ET are co-funded by the Medical Research Council of the UK (grant number MR/P027873/1) through the Global Challenges Research Fund. However, the funders had no role in study design, data collection and analysis, decision to publish, or preparation of the manuscript.

**Competing interests:** We have read the journal's policy and an author of this manuscript have the following competing interests: EDS holds a position funded by Vestergaard Sarl. The other authors declare no competing interests.

Recent research has highlighted the potential of housing improvement [3–5] including novel housed-based interventions for insecticide delivery [6–9]. An example is the eave tube, [10,11] which involves fitting pieces of PVC tubing to otherwise closed eaves, to act like funnels to channel human host odours out of the house. As host-searching mosquitoes recruit to the tubes they encounter an insert that serves to both block entry and deliver an insecticide. One version of this approach is the In2Care® Eave Tubes, in which the insert comprises netting treated with an electrostatic coating that enables use of powder formulations of insecticide [12,13].

A range of laboratory and semi-field studies indicate that Eave Tubes can reduce survival of host-searching mosquitoes as they attempt to enter the house, including those mosquitoes exhibiting high levels of insecticide resistance [10,14,15]. Most recently, a two-year cluster randomized controlled trial (CRT) conducted in 40 villages in central Côte d'Ivoire [13,16] has shown that household screening (which included screening of windows, closing of any open eaves, fixing of other large gaps in the walls, etc.) combined with Eave Tubes can provide protection against malaria and is similarly cost-effective to other core vector control interventions [16]. The CRT demonstrated an overall reduction in incidence of clinical malaria in cohorts of children in the treatment villages of 38% compared with the control villages. In villages where coverage of the intervention was above 70% (13 out of the 20 in the intervention arm), incidence of clinical malaria was 47% lower than in control village clusters [16].

Given the apparent promise of Screening + Eave Tubes (SET), one of the priorities is to determine the potential to scale-up the approach to operational levels and to identify those areas that are most suitable. One possible limitation of SET is the need for the house to be constructed of appropriate materials to enable efficient installation; it can be difficult to block the eaves in houses with mud walls and/or thatched roofs, and mud-walled houses might not be sufficiently robust to allow fabrication of holes to hold the tubes. However, in the past decade, there has been an intense urbanization and house improvement in sub-Saharan Africa, [17] and improved housing is likely to support the implementation of SET.

We created a geospatial model to predict the suitability for SET across Côte d'Ivoire to identify priority areas for additional ground truthing and calculate the potential population coverage and costs prior to implementation.

## Materials and methods

### Study site

Côte d'Ivoire is in West Africa; the political capital is Yamoussoukro and the economic capital Abidjan. Malaria cases are mostly caused by *Plasmodium falciparum* and in 2019, the incidence was 190 per 1000 in the population and 493 per 1000 in children under five [18]. The parasite prevalence varies from ~60% in the West and South-West districts, to 3% in the city of Abidjan (Fig 1). The main vectors are *Anopheles gambiae s.s.*, *Anopheles coluzzii*, and *Anopheles funestus s.s.* The major vector, *An. gambiae s.l.*, is resistant across the country to the most commonly used insecticides [18].

### Data sources and analysis

We extracted details of roof and wall materials from 9686 sampled households in the 2011–12 Demographic Health survey (DHS) [19], and classified them as appropriate or inappropriate for SET [13,16]. Appropriate wall materials included cement, bricks, cement blocks, and stone with lime/cement. Appropriate roof materials included metal, cement, ceramic tiles, and roofing shingles (S1 Table in S1 File). We considered a household to be suitable for SET when both roof and wall materials were appropriate.

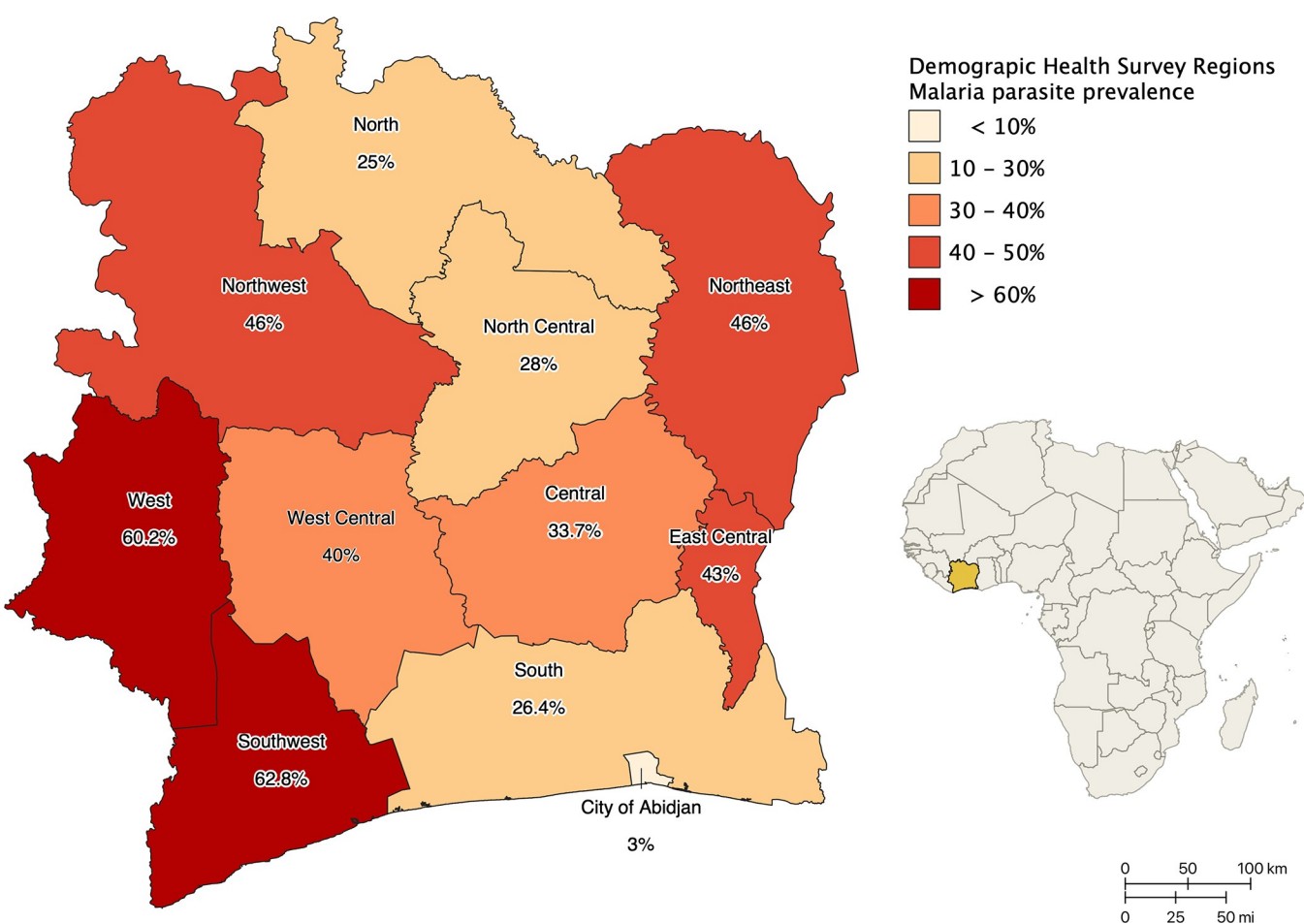

**Fig 1. Malaria parasite prevalence by microscopy in children 6–59 months in the districts of Côte d'Ivoire, 2016.** Data source [18]. Basemap: https://data.humdata.org/dataset/cote-d-ivoire-administrative-level-0-3-boundaries-and-points.

Individual household-level DHS survey data were aggregated into the 341 DHS clusters, which represent groups of households that participated in the survey. Cluster size ranged from 15 to 32 households. For privacy reasons, DHS randomly displaces the cluster coordinates from their true location by up to 2 km in urban and up to 10 km in rural areas [20]. Due to the possible large displacement of rural clusters, we re-georeferenced them following the methods of Grace et al. (2019) [21] (S2 Table and S2 Fig in S1 File) which involves using satellite images to identify the nearest populated rural location to the DHS cluster and adjusting the location to match that of the nearest populated rural setting. Urban clusters were not adjusted.

Descriptive analysis of household suitability within clusters was done by calculating the DHS cluster suitability (% of houses with appropriate materials within a cluster). This data was aggregated within districts to explore spatial heterogeneity.

## Geostatistical model

The geostatistical model used covariates previously identified to have a significant relationship with housing development [3]: aridity [22]; time to the nearest city [23]; degree of urbanisation [24]; and value of production of food and non-food crops [25] (S3 Table and S3 Fig in S3 File). For each cluster, we created a 2 km buffer to extract and calculate the average raster value for each covariate.

To produce the geostatistical model, we fitted a Bayesian beta-binomial logistic model using the integrated nested Laplace approximation (INLA). We accounted for the spatial correlation using stochastic partial differential equation (SPDE) models, generated from a Delaunay triangulation mesh of our study region to estimate a continuous Gaussian field and provide a fast approximation of the posterior marginal distribution when compared to other Markov chain Monte Carlo (MCMC) approaches [26,27].

The beta-binomial distribution is characterised by having the probability parameter for the binomial distribution $P(x_i)$ drawn from a Beta prior [28]. The proportion of suitable houses was modelled using the spatial covariates $(B_j(x_i))$ and a zero-mean Gaussian process with a Matern kernel $(S(x_i))$.

$$Y_i|P(x_i) \sim Bin(n, P(x_i))$$

$$P(x_i) \sim Beta(\alpha, \beta)$$

$$logit(P(x_i)) = \beta_0 + \beta_j(x_i) + S(x_i)$$

We estimated the model coefficients and their associated 95% confidence intervals using the spatial covariates outlined above. For each 5 km$^2$ grid square, the primary model outputs were 1) a point estimate for the proportion of houses that are suitable, and 2) an exceedance probability of attaining a threshold of 80% suitable households, based on the inclusion criteria of the Côte d'Ivoire SET CRT [13] in which ≥80% of the houses in the study villages had to be suitable for modifications with SET. The trial results [16] also suggest community protection by reducing risk of malaria case by 47% when SET coverage is greater than 70%. With less than 70% the trial also suggests some protection.

Due to the out-dated DHS household data, we performed a simulating scenario by calculating an improvement of 20% on roof and walls materials to see the effects on the estimates. Tusting et al. 2019 [3] calculated the changes in households in SSA from 2000 to 2015. The 20% is roughly the improvement between the years 2000 and 2015. Although part of this 20% is captured in our data we maintain this rate to account for the years after 2015.

## Priority areas, target population and cost

To identify priority areas for ground-truthing of our geostatistical model output and to inform subsequent SET implementation, we excluded 5 km$^2$ grid squares with (Fig 2):

1. an exceedance probability of less than 70% of reaching the household suitability threshold, as estimated from the geostatistical model described above.

2. low malaria transmission (<10% parasite rate) as a proxy for potential epidemiological impact.

3. >3 hours travel time from a major city as a proxy for implementation practicality [23].

This resulted in a set of included 5 km$^2$ grid squares deemed suitable for implementation of SET, which were subsequently used to calculate the targeted population (using data from [29]) and estimate the cost of a strategy focussing either on targeting only high transmission areas (>35% malaria parasite rate) or, one targeting moderate and high transmission (>10% malaria parasite rate) areas. For each strategy, total cost was estimated by multiplying the estimated number of people protected by the estimated cost per person protected per year from the previous CRT [16], $21·47 (90% credible interval $6·08 to $49·99).

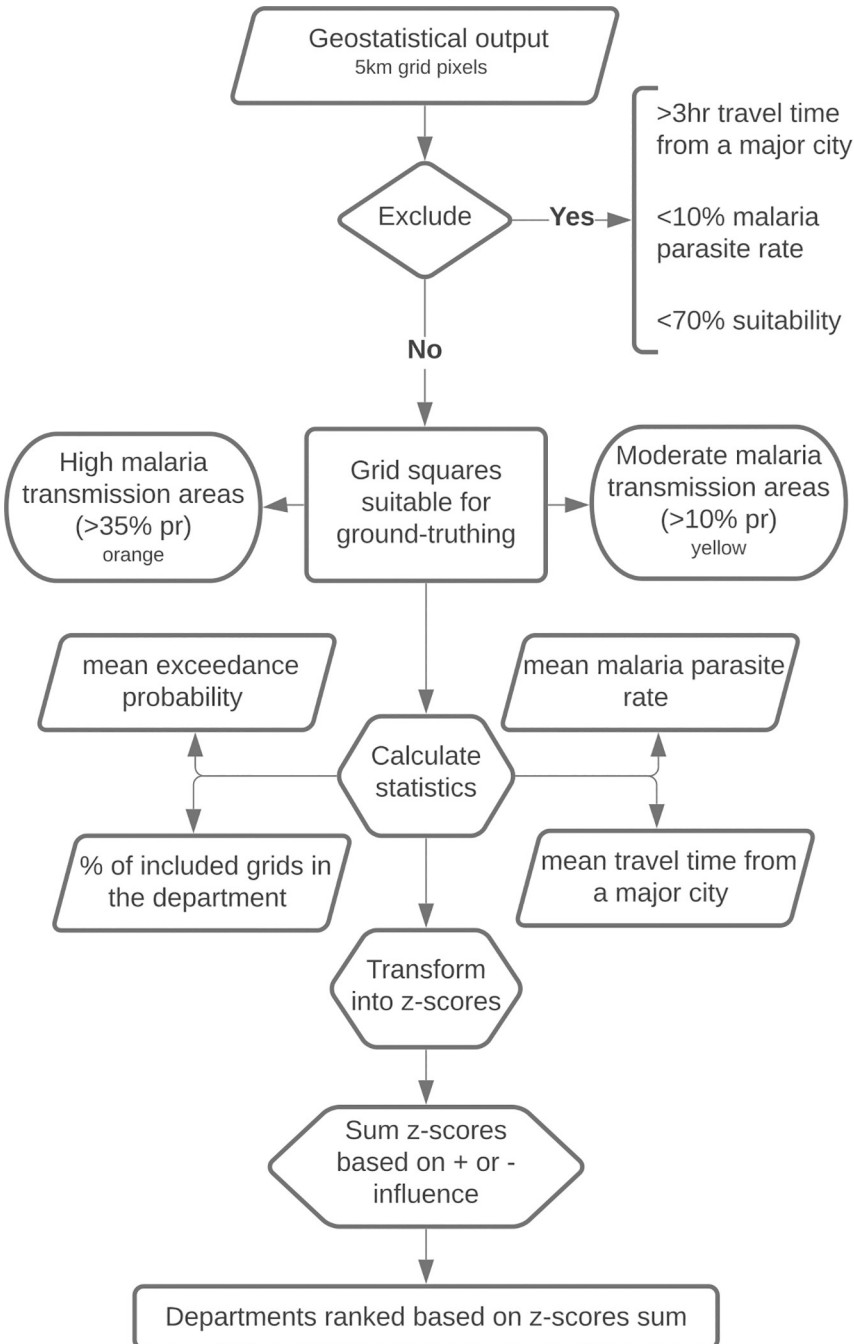

**Fig 2. Flow chart to select the grid squares for the ground-truthing and to rank departments into ground-truthing/implementation priority.**

A 5 km$^2$ grid is not a meaningful spatial scale with which to make implementation and resource prioritisation decisions. We therefore conducted a ranking exercise to identify and prioritise departments (admin level 3 in Cote d'Ivoire) (Fig 2):

1. We calculated the mean exceedance probability, mean malaria parasite rate, and mean time to the nearest city for each included grid square, department, and the entire country

(national). We also calculated the percent of the total area of each department that comprised included grid squares and the national mean of included area per department.

2. We transformed the mean values for each variable (exceedance probability, parasite rate, time to nearest city and included area) into z-scores, using the mean value for the department as our observed value, and the national mean as the population average.

3. The z-scores for each department were then summed based on their positive or negative influence to prioritize SET. Departments were then ranked based on the highest summed z-score value.

Maps were generated in QGIS (v. 3.16). All statistical analyses were carried out in R (v. 4.0.3), and the geostatistical model was generated using the INLA package. Covariates, model output and ranked priority areas can be visualised using an online mapping tool (R shiny app) at https://et-ivc.shinyapps.io/Shiny/.

## Results

### DHS cluster suitability

There was heterogeneity in cluster suitability across the country (Fig 3). The urbanised districts of Abidjan and Yamoussoukro showed highest suitability (Fig 3B). Most of the variation was attributable to urban/rural classification; urban clusters were often above the 80% household suitability threshold, while rural clusters were below (Fig 3C and 3D). Materials used for house construction differed in urban and rural locations: houses in rural areas most often had mud walls while those in urban locations used cement (S4 Fig in S1 File).

### Model predictions of suitability

The main cities of Abidjan, Bouake, and Yamoussoukro and their surroundings have a high probability of exceeding the 80% suitability threshold (Fig 4). In proximity to these cities are scattered areas of high suitability in satellite towns and villages. The 5 km$^2$ grid squares in the west and east parts of the country have very low exceedance probabilities. The large low-exceedance probability areas in the southwest and northeast can be explained by the Taï and Comoé national parks, respectively. The model outputs of suitability and standard errors are available in S5 Fig in S5 File.

### Suitable areas and SET cost

We identified several areas throughout the country as suitable areas for ground-truthing based on exceedance probability, malaria burden, and accessibility (yellow and orange areas of Fig 5). Overlaying the suitable areas to the population distribution, we estimate that SET could potentially protect 31% of the total Côte d'Ivoire population, and the same amount of the population already living in moderate or high transmission areas. If SET was only implemented in high transmission areas, it could protect 8% of the total national population, and 17·5% of the people already living in the high malaria transmission areas. Most suitable areas in the high malaria transmission are in the middle of the country, around the cities of Yamoussoukro and Bouaké, where the SET CRT was done [16]. The capital, Abidjan, is the largest suitable area in moderate transmission but represents the largest amount of population that could be protected.

The estimated cost of implementing SET in the moderate and high transmission suitable area (yellow and orange in Fig 5) would be in the region of $174 million ($49m - $405m) and will cover around 8M inhabitants. For the high malaria transmission areas (orange), it would

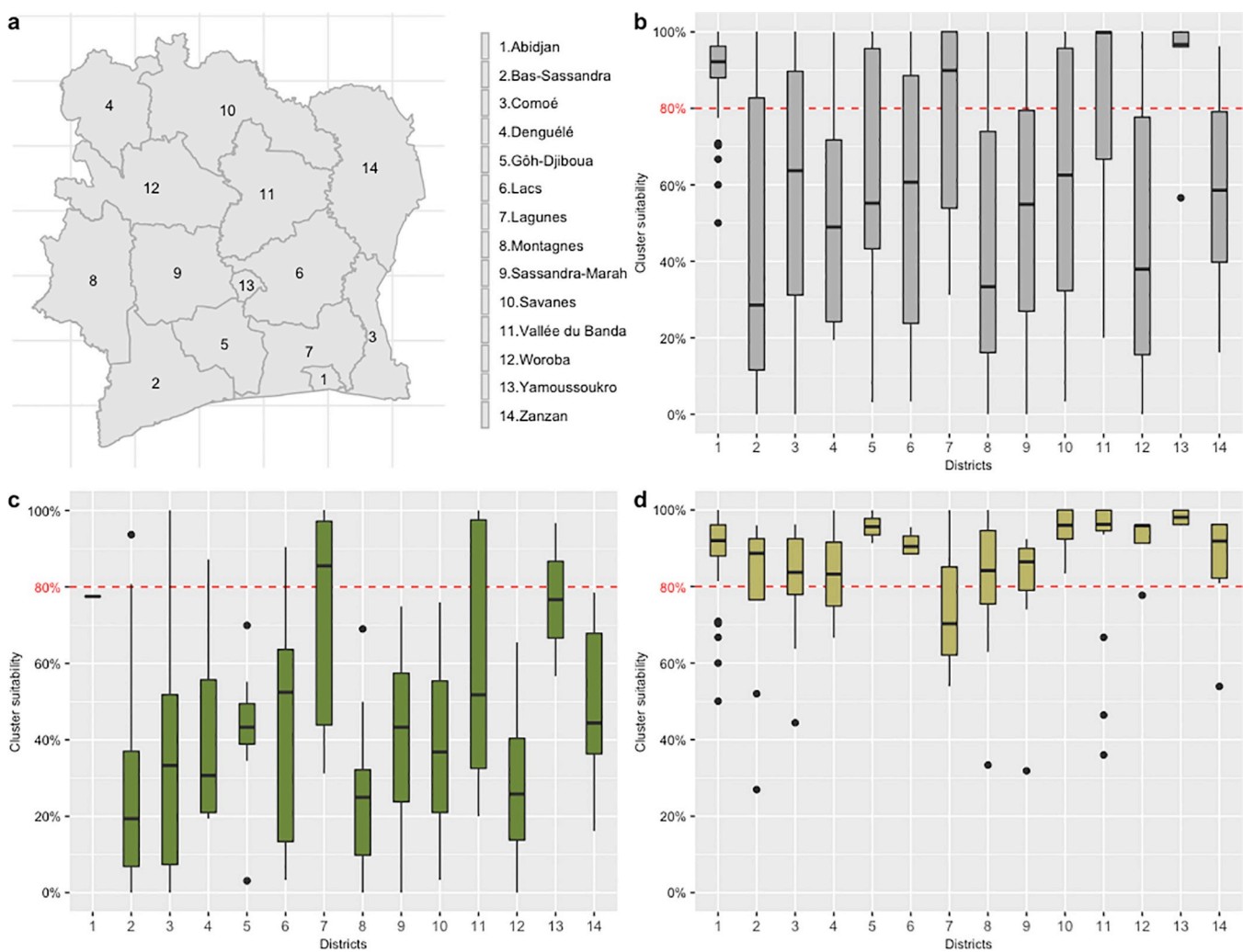

**Fig 3.** a-District locations; b-Rural and Urban clusters; c-Rural clusters; d-Urban clusters. The boxplots represent the cluster suitability (% of households with appropriate materials within a cluster) aggregated into districts. Middle lines are the median, box ranges from first to the third quartile, and whiskers indicates min and max. Black filled circles are outliers. The red dotted line marks the 80% desired suitability for the intervention. Basemap: https://data.humdata.org/dataset/cote-d-ivoire-administrative-level-0-3-boundaries-and-points.

cost $46m ($13m –$108m) and a cover up of 2M. Further ranking by departments revealed that nine out of the top ten departments are in the center of the country. These are all departments with high malaria transmission, and their main cities and towns are relatively well connected. Abidjan is the only top-ranked department outside the center and the high malaria transmission area. This can be explained by high exceedance probabilities and accessibility. The full priority list is provided in S6 Table in S6 File.

To explore the feasibility of financing SET, we compared estimated SET implementation costs (moderate and high transmission and high transmission only strategy) with relevant malaria program budgets as well as with the annual budget of the Côte d'Ivoire Ministry of Health (MoH) (Fig 6). Supporting either of these strategies from existing malaria and health budgets is clearly unaffordable, as is the case with Indoor Residual Spraying (IRS) programme supported by the US Government Presidents Malaria Initiative (PMI). IRS (which theoretically would be applicable in all homes across Côte d'Ivoire) is therefore targeted to two departments (Sakassou and Nassian) within Côte d'Ivoire. These two departments represent 0·6% of the

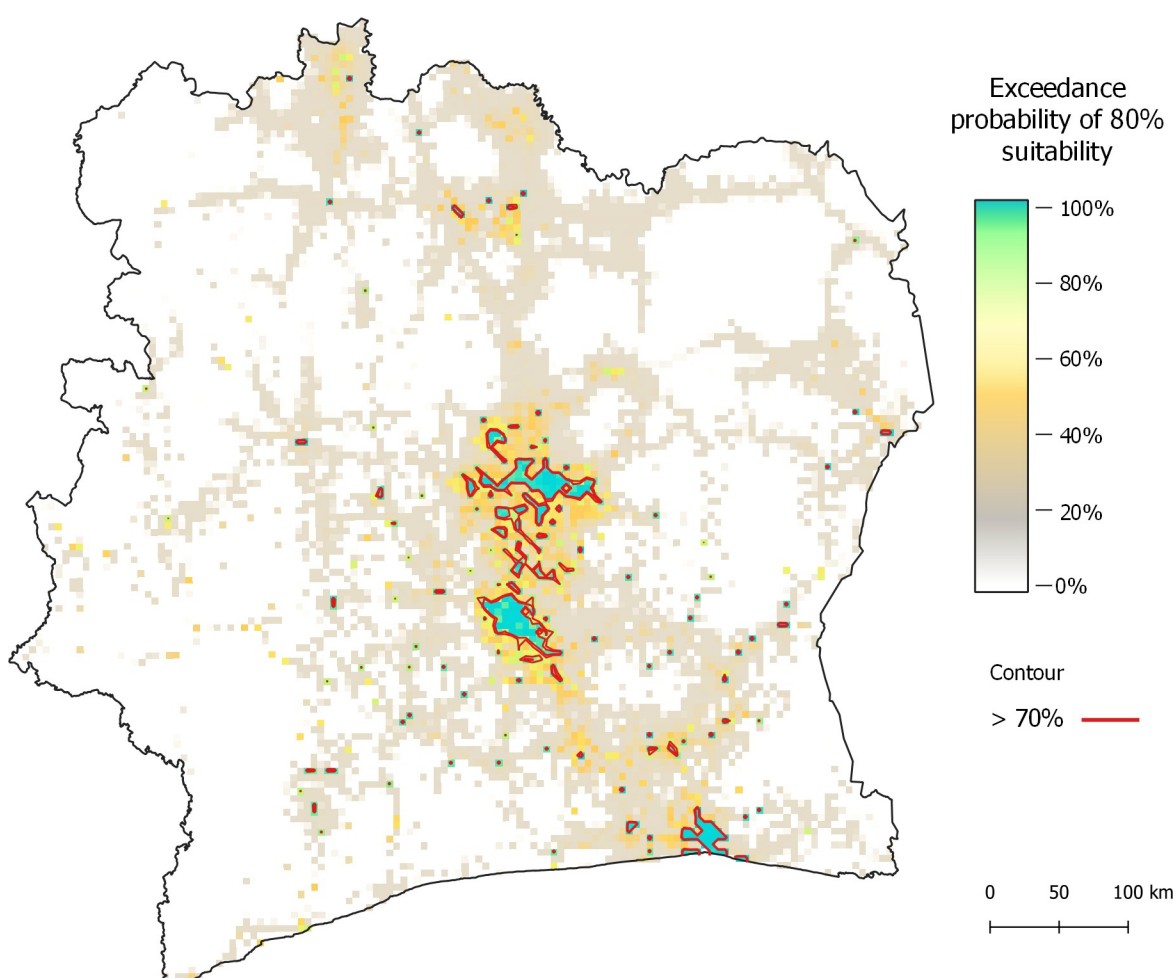

**Fig 4. Probability of a 5 km² grid square exceeding the threshold of 80% suitability.** Red contour line delineates probability greater than 70%. Basemap: https://data.humdata.org/dataset/cote-d-ivoire-administrative-level-0-3-boundaries-and-points.

national population, compared to 31% (high and medium) and 8% (high only) of the national population that would be reached in our modelled SET scenarios. Widespread health financing constraints even for cost-effective interventions such as IRS and SET, highlight the importance of the department level prioritisation exercise conducted in this paper. Furthermore, we identified a donor-funded housing project in Côte d'Ivoire with a larger budget than the amount required to implement SET in high transmission settings (Fig 6), revealing the need for innovative financing strategies to support interventions like SET which support multiple sustainable development goals.

## Discussion

The WHO Global Technical Strategy advocates for spatial targeting of new malaria vector control tools [1]. However, few examples of how to do this exist. Here we develop a nation-wide model for SET, highlighting the potential protection of 31% of the national population, and the population already living in moderate to high transmission areas. The scenario created provides departments and areas where ground-truthing is a priority for future research to support eventual implementation. Further, we calculated the cost of implementing the

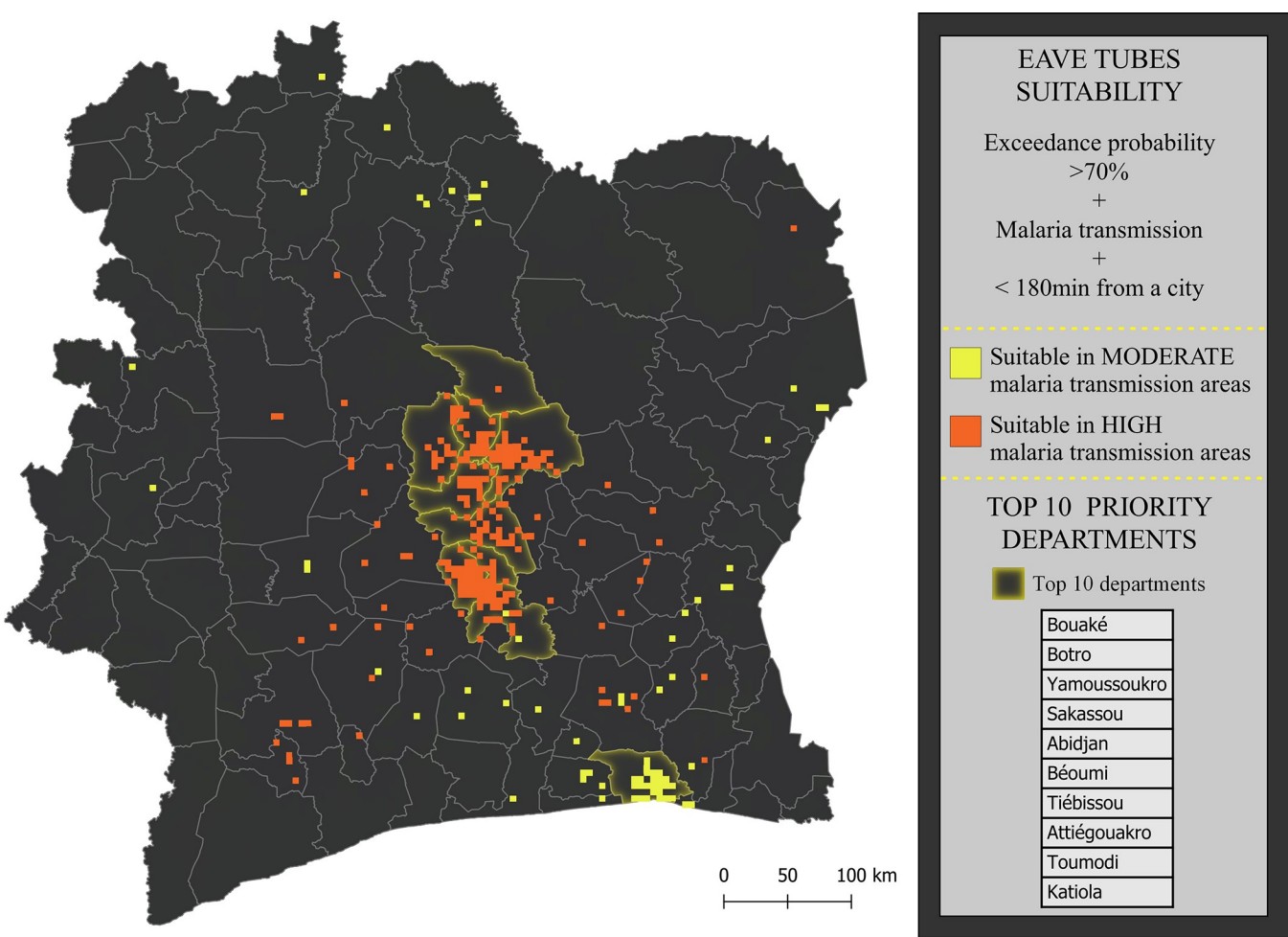

**Fig 5. Priority areas for further ground-truthing in preparation for SET implementation prioritization and planning.** Suitable areas (orange and yellow 5 km² squares) were identified by excluding areas with low malaria transmission (<10% parasite rate), low accessibility (>180 min from nearest city) or poor household suitability for SET (exceedance probability of >70% from model output). Departments highlighted in yellow are the top ten priority departments after ranking them according to the exceedance probability, malaria transmission, and accessibility of grid squares within their borders. Basemap: https://data. humdata.org/dataset/cote-d-ivoire-administrative-level-0-3-boundaries-and-points.

intervention at scale, explored financing feasibility within existing malaria and health budgets, and provided suggestions for prioritisation and innovative financing options.

A major limitation of this study is that the underlying housing data were from 2011. There has been rapid rural development in sub-Saharan Africa in the last decade [3,35], which these data do not capture. For this, our simulating scenario (S6 Fig in S6 File) reveals an increase of suitable areas in rural areas, and scattered villages and towns become highly suitable nation-wide. Between Abidjan and the centre of the country there is almost a corridor of suitability. Notably there is a big increase in suitability in two northern cities Korhogo and Tingrela, revealing an area that with the current analysis is totally discarded. Although theoretical, this analysis helps to pinpoint areas with the most potential to be suitable.

The results highlight the importance of urban and peri-urban areas for the implementation of SET. Peri-urban areas are growing rapidly due to the increase in urban populations, attracted by the economic situation of main urban centres [36]. For example, the expansion of Abidjan was 4% in the last ten years [37]. Peri-urban and urban environments benefit

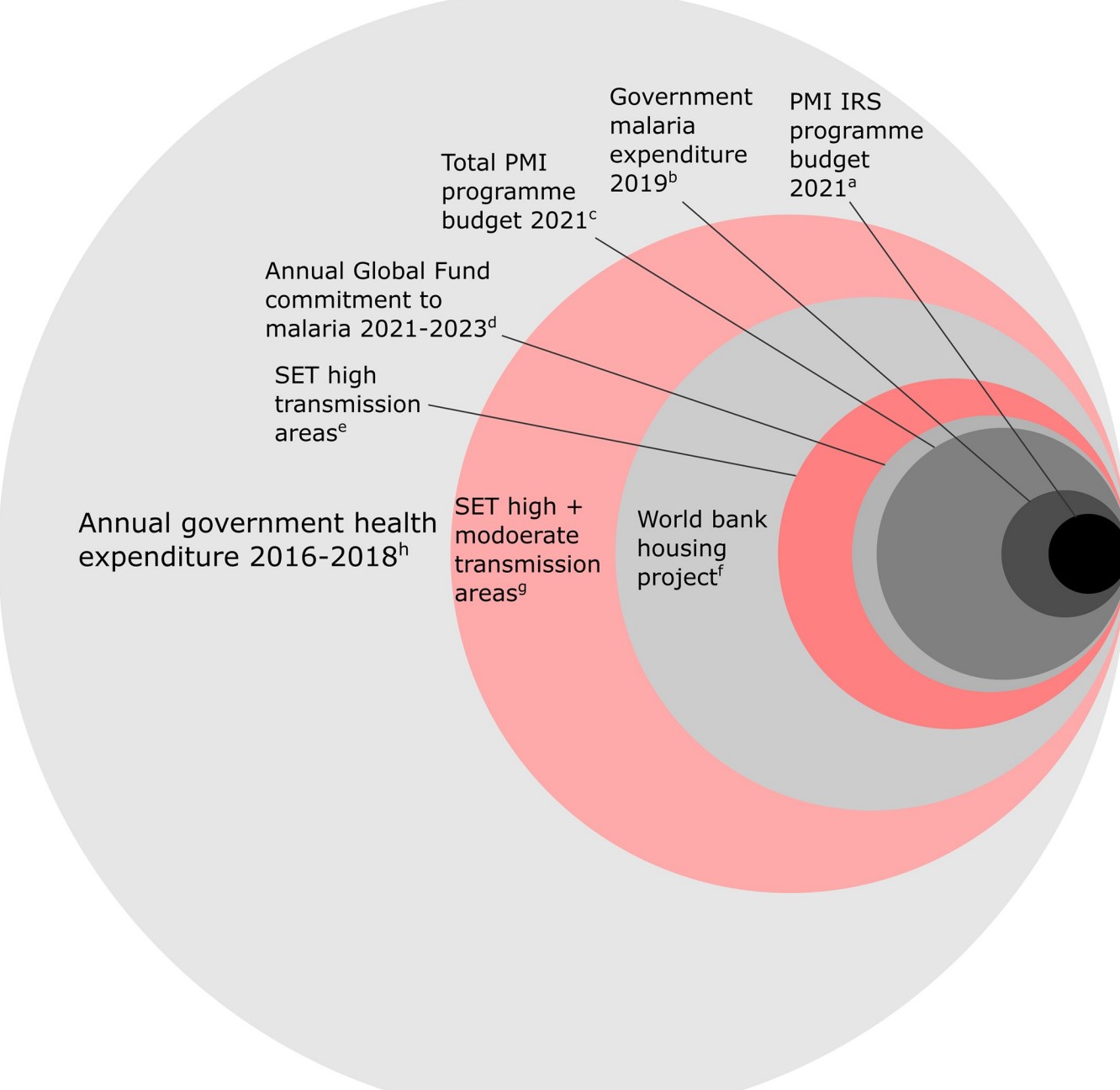

**Fig 6. Comparison between estimated cost of SET in high and/or moderate malaria transmission areas of Côte d'Ivoire with example budgets.** [a] US Government Presidents Malaria Initiative, planned obligation (budget) for Indoor Residual Spraying in Cote d'Ivoire for financial year 2021 ($2.4 M) [30]; [b] Domestic government malaria control expenditure, Cote d'Ivoire (2019) ($6 M) [31]; [c] US Government Presidents Malaria Initiative, planned obligation (budget) for all Malaria activities in Cote d'Ivoire for financial year 2021 ($24 M) [30]; [d] Global Fund Scaling up of interventions to combat malaria in Côte d'Ivoire grant phase 3 award (2021 to 2023), total commitment amount divided by three $86.756.101/3 [32]; [e] Annual cost of SET in high transmission areas ($46 M); [f] International Finance Corporation, World Bank group providing $100m investment to support construction and commercialisation of affordable housing in Côte d'Ivoire [33]; [g] Annual cost of SET in moderate + high transmission areas ($174 M); [h] Average annual domestic government health expenditure, Cote d'Ivoire MoH 2016–2018 ($ 484 M) [34].

logistically by being well-connected and encompass an increasing proportion of the population. In addition, improving housing conditions (by screening and closing eaves) can improve comfort of the inhabitants and could potentially provide protection against other pathogens such as arboviruses that have increased alarmingly in Côte d'Ivoire [38,39]. However, this rapid growth and high population can lead to proliferation of slums comprising low quality housing [36]. The SET CRT has not explored these areas [13] and therefore we do not know how effective the intervention could be in such settings. This should be a priority for future research.

Rapid urbanisation creates substantial opportunity for housing modification and improvements, with the potential to tackle public health problems beyond malaria [40–42]. However, high heterogeneity in house construction and design, as well as complexity in issues of house and land ownership, likely create additional challenges for implementation of house-based interventions in dense urban settings. One part of this challenge is ensuring high enough coverage (i.e., proportion of structures receiving the intervention) to provide community benefit. The results of the CRT [16] suggest that there can be community benefit of SET when coverage exceeds 60–70%. This community benefit is important in terms of equity as it potentially reduces transmission risk in households that do not receive the intervention. Also, studies conducted in individual experimental houses within the CRT villages suggest that while screening can provide some protection against indoor biting mosquitoes at household level, Eave Tubes appear to provide little direct benefit, either when used with screening, or as a standalone tool [43]. That said, the mode of action of Eave Tubes is to kill mosquitoes outside the house and as with IRS, this indirect effect is only likely to be beneficial in impacting transmission if coverage is high enough to lead to reductions in mosquito density or to changes in population age structure.

Our cost estimates indicate that large-scale adoption of housing improvement does not come cheap in absolute terms. However, there are several factors to consider in this context. First, the costs we use for SET derive from an initial experimental field trial that served as proof of principle. It is likely that more extensive area-wide adoption would lead to economies of scale in production, distribution, and implementation. Benchmarking against expenditure on other large-scale donor financed malaria control interventions (Fig 6) illustrates the large sums of money that are dedicated to addressing malaria. With political will, funds follow effective interventions (as in the case of IRS and artemisinin-based combination therapy (ACTs)), and the recent CRT, which suggests SET to be similar in cost-effectiveness to certain other core vector control tools, should help generate political and financial support for this new tool from existing and novel partners [16]. Second, even though implementation of SET might be motivated by the desire to reduce malaria burden, it does not necessarily follow that this expenditure needs to be wholly supported by limited public finances. To build a modest house with contemporary materials in one of the CRT study villages costs around $6000 [43] and householders already spend money on housing improvement (~$400 for doors and windows) creating the potential to leverage private investment for public health benefit. Benchmarking against the sums of donor assistance available for public private investment in support of the Sustainable Development Goals illustrate the huge potential to tap into vast sums [44]. Third, households are being built with progressively better materials [3] and integrating Eave Tubes and other housing improvements such as screening and blocking eaves as part of the new-build, could reduce costs compared with retro-fitting modifications to existing structures [41,45]. However, developing guidance frameworks for housing design, and possible incentive or regulatory mechanisms to facilitate adoption, would require close collaboration between the health and housing sectors through interministerial working groups [2], and strong support for the intervention at the highest levels of government. Geospatial modelling of the type

presented here could contribute to initiating this dialogue and serve to guide any deployment pilots in the interim.

## Conclusion

The implementation of Screening+Eave Tubes (SET) in Côte d'Ivoire is currently limited to the villages associated with the initial CRT [13]. The results of the CRT are encouraging [16] but for the approach to have greater impact and relevance it needs to be scalable. The current study provides insight into where such scale-up might be most effectively targeted and its approximate cost. Overall, we find that the most suitable areas are in urban and peri-urban environments. Inclusion of more up-to-date data on housing construction would likely increase the number of houses and the total area predicted to be suitable for the intervention. Nonetheless, even with the current data the suitable areas cover 17·5% and 31% of the population who are living in areas categorized with high and moderate malaria transmission. This finding highlights the importance of evaluating the efficacy, acceptability and suitability of SET in peri-urban environments, as well operational research to optimise implementation in these settings. Reciprocally, by highlighting areas where SET is less appropriate, the study also provides insights into where alternative control tools need to be deployed, contributing to the goal of targeted, integrated control strategies [1].

Housing modification for vector control has been re-introduced on the global agenda [46,47]. Leveraging housing improvement offers a key opportunity to boost malaria control by accessing public and private sector funding beyond the vertical donor funded mechanisms that currently support public health [48]. It has been estimated that homes for up to 2 billion people will need to be built by 2050 to accommodate the growing population in Africa, which is equivalent to building housing for 7000 people/hour for the next 30 years. We have an opportunity to guide this development to deliver co-benefits of improved housing and reductions in vector-borne disease. Our study demonstrates a robust method of using publicly available data to inform spatial targeting of house-based interventions for maximum impact. As evidence for the effectiveness of SET or other house-based interventions grows, this same methodology could be used in other settings and even on a continent-wide scale across sub-Saharan Africa.

## Supporting information

**S1 File.**
(DOCX)

**S2 File.**
(DOCX)

**S3 File.**
(DOCX)

**S4 File.**
(DOCX)

**S5 File.**
(DOCX)

**S6 File.**
(DOCX)

## Author Contributions

**Conceptualization:** Eve Worrall.

**Data curation:** Tiago Canelas, Daniel McDermott.

**Formal analysis:** Tiago Canelas, Edward Thomsen, Daniel McDermott, Eve Worrall.

**Funding acquisition:** Matthew B. Thomas, Eve Worrall.

**Investigation:** Tiago Canelas, Edward Thomsen, Eleanore Sternberg, Matthew B. Thomas, Eve Worrall.

**Methodology:** Tiago Canelas, Edward Thomsen, Daniel McDermott, Eleanore Sternberg, Eve Worrall.

**Resources:** Matthew B. Thomas.

**Software:** Tiago Canelas, Daniel McDermott.

**Supervision:** Tiago Canelas, Edward Thomsen, Matthew B. Thomas, Eve Worrall.

**Validation:** Tiago Canelas, Edward Thomsen, Eleanore Sternberg, Matthew B. Thomas, Eve Worrall.

**Visualization:** Tiago Canelas, Eve Worrall.

**Writing – original draft:** Tiago Canelas.

**Writing – review & editing:** Tiago Canelas, Edward Thomsen, Daniel McDermott, Eleanore Sternberg, Matthew B. Thomas, Eve Worrall.

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
