## [Decision Letter · Decision Letter 0]

20 Jul 2021

 PGPH-D-21-00253 Spatial targeting of a house-based malaria control intervention PLOS Global Public Health

Dear Dr. Canelas,

Thank you for submitting your manuscript to PLOS Global Public Health. After careful consideration, we feel that it has merit but does not fully meet PLOS Global Public Health’s publication criteria as it currently stands. Therefore, we invite you to submit a revised version of the manuscript that addresses the points raised during the review process.

We look forward to receiving your revised manuscript.

Kind regards,

Louisa Alexandra Messenger, MSc, PhD

Academic Editor

Journal Requirements:

Additional Editor Comments (if provided):

Reviewers' comments:

Reviewer's Responses to Questions

**Comments to the Author**

1. Does this manuscript meet PLOS Global Public Health’s publication criteria? Is the manuscript technically sound, and do the data support the conclusions? The manuscript must describe methodologically and ethically rigorous research with conclusions that are appropriately drawn based on the data presented.

Reviewer #1: Yes

Reviewer #2: Yes

2. Has the statistical analysis been performed appropriately and rigorously?

Reviewer #1: Yes

Reviewer #2: Yes

3. Have the authors made all data underlying the findings in their manuscript fully available (please refer to the Data Availability Statement at the start of the manuscript PDF file)?

Reviewer #1: Yes

Reviewer #2: Yes

4. Is the manuscript presented in an intelligible fashion and written in standard English?

Reviewer #1: Yes

Reviewer #2: Yes

5. Review Comments to the Author

Reviewer #1: General comments:

In this study, the authors used a geostatistical model to predict household suitability for Screening + Eave Tubes (SET) in Cote d'Ivoire (defined based on housing materials), and then used the model output alongside other factors to identify priority areas for ground-truthing for potential SET implementation. The study was based on 2011 DHS data from Cote d’Ivoire, which are the latest publicly available national-level georeferenced data. This study addresses the important issue of local adaptation and targeting of interventions for malaria control and elimination, and it provides a data-driven approach to identifying locations that could be prioritized for implementation of a promising house-based intervention. It also provides estimates of cost and highlights the need for financing strategies to bring SET interventions to scale. The article is well-written and methods are well explained, however some aspects could be further clarified.

Specific comments:

1. A more informative title with location and study type would be useful; E.g. “Spatial targeting of Screening + Eave Tubes (SET), a house-based malaria control intervention, in Cote d’Ivoire: a geostatistical modelling study”

2. Abstract: the objective statement indicates the study aims to “define the most appropriate locations where Screening + Eave Tubes (SET) could be implemented”, but this is a bit misleading since the intent is not to directly inform intervention implementation, rather it is a proof of principle study to generate estimates that would require further validation prior to any implementation. Suggest modifying the abstract to match the objective statement in the introduction, i.e. “identify priority areas for ground-truthing of geostatistical model output”.

3. In the abstract, more information is needed on how “suitability of SET” and “impact” are defined.

4. Introduction: here it is apparent that suitability refers to the presence of improved house construction, but “impact” is not described in the introduction and should be described here if it is an objective.

5. Methods, line 77: Some rationale for using DHS data from 2011 should be presented in the methods, considering that the information is ten years old, and does not accurately reflect the current status of housing materials in many areas. Indeed, the introduction mentions that “in the past decade, there has been an intense urbanization and house improvement in sub-Saharan Africa”. Did the authors request GPS data from the 2016 MICS or perform any other kind of validation to assess data accuracy? This is a limitation of the study that affects all estimates derived from the model output, therefore a more cautious tone would be appropriate to frame the study as a proof of principle (see comment #8 and #10 below).

6. Methods, line 79-80: it would be useful to define “appropriate” in the body of the paper, especially since this definition essentially delineates between urban and rural houses (as mentioned in the results lines 157-159).

7. Methods: the authors describe a sequence of data processing steps that they followed to calculate the target population and cost, based on the gridded model output. They then derive department-level variables from the model output (exceedance probability and included area) and combine these with department-level data on malaria parasite rate and travel time to produce a ranking of departments to prioritize for ground-truthing based on summed z-scores. The approach that was used appears sound, however there is a high likelihood of introducing error at each step, making the end results less robust. A figure showing the data processing steps (e.g. flow chart) would be useful so that it is clear which data were aggregated or combined for which outcome measure.

8. Methods: Did the authors consider conducting a sensitivity analysis to see what changes would result from variability in the model input measures – for example, if it was assumed that 20% (or perhaps more, depending on the rate of urbanization) of houses in each cluster were improved between 2011 and 2021 how would this affect the estimates? Presumably more areas would meet the exceedance threshold than were identified by the initial model.

9. Results, line 216-218: is there a word missing between ‘theoretically’ and ‘SET’?

10. Discussion: the limitations of the 2011 DHS data are explicitly mentioned, but it is not clear what is meant by “To account for the out-to-date housing data and the influence of the covariates in the model, we created an operational scenario to help guide and target the implementation or future trials”.

Reviewer #2: SET has demonstrated substantial public health benefit in a randomised control trial in Cote d’Ivoire though it is unclear where it can be implemented given the requirement of concrete housing. This manuscript sets out a geostatistical method of assessing where it might be useful to introduce and estimates the cost. There is a need for this type of work and the manuscript uses sensible geospatial methods. The paper is well written and is generally well written, though I fear the resolution of these data and the ecological fallacy makes me question some of the results.

1). The major problem as I see it is one of geographical scale. You are looking at an intervention which requires concrete housing which are more likely in urban areas. It is not mentioned in the manuscript but it is widely understood thar malaria is less common in heavily urbanised area due to the lack of availability of suitable mosquito breeding sites. Therefore areas most suitable to SET might have the least need for it from a malaria control perspective. Malaria data is taken from DHS data summaries at a very high geographical scale (admin1) and as i understand it DHS is generally collected in rural areas. Therefore the suitability of the admin 3 is driven by this crude approximation of whether the location is of moderate or high transmission. In the crudest scenario you might have an admin 3 unit with a large urban area with all concrete houses but no malaria and a small rural area with lots of malaria but no concrete houses. The site of the CRTC was clearly high malaria and widespread concrete housing but this is unlikely to be universal, suggesting the numbers who could be protected and the costs could be exaggerated. This does not invalidate the models identifying areas with concrete housing, but the authors might want to consider using more fine resolution outputs from the MAP to identify high malaria areas or perhaps using maps of rural/urban breakdown to exclude those urban areas in a sensitivity analyses. Either way the issue needs to be highlighted.

2). Title seems inappropriate. It could be argued that LLINs and IRS are house-based malaria control and certainly there are other housing improvements that could be implemented in houses unsuitable for SET. Consider revising.

3). Line 21 of the abstract says that the work “and to calculate the impact” of SET, but this does not appear to be the case.

Minor points

1). Language needs to be tighter. In the abstract as it says “SET could protect 31% of the total population and 17·5% of the population in areas of high malaria transmission”. As I understand it 31% of structures people live in the could implement SET. Other people in these communities could benefit from SET (as is mentioned) so saying just those using it are protected is a little misleading. Similarly, you are talking about high transmission admin 1 units (districts?) and this should be stated here as there will be other high transmission areas within low transmission districts.

2). Latin names of mosquitoes should be written in full the first time they are used (even if the genus has been abbreviated before, eg. Line 70).

3). Line 117 says that “The trial results [16] also suggest community protection is achieved when ≥70% of households have SET”. This indicated that community protection is either on or off, when it has been established that it is more of a continuum. Consider changing.

4). Line 208 typo “Supporting the either….”

6. PLOS authors have the option to publish the peer review history of their article (what does this mean?). If published, this will include your full peer review and any attached files.

**Do you want your identity to be public for this peer review?** For information about this choice, including consent withdrawal, please see our Privacy Policy.

Reviewer #1: No

Reviewer #2: No

---

## [Editor Report · Decision Letter 1]

18 Oct 2021

Spatial targeting of Screening + Eave tubes (SET), a house-based malaria control intervention, in Côte d’Ivoire: a geostatistical modelling study

PGPH-D-21-00253R1

Dear Dr. Canelas,

We're pleased to inform you that your manuscript has been judged scientifically suitable for publication and will be formally accepted for publication once it meets all outstanding technical requirements.

Within one week, you'll receive an e-mail detailing the required amendments. When these have been addressed, you'll receive a formal acceptance letter and your manuscript will be scheduled for publication.

An invoice for payment will follow shortly after the formal acceptance. To ensure an efficient process, please log into Editorial Manager at https://www.editorialmanager.com/pgph/ click the 'Update My Information' link at the top of the page, and double check that your user information is up-to-date. If you have any billing related questions, please contact our Author Billing department directly at authorbilling@plos.org.

Kind regards,

Louisa Alexandra Messenger, MSc, PhD

Academic Editor